# Prognostic Value of Routine Biomarkers in the Early Stage of COVID-19

**DOI:** 10.3390/healthcare11152137

**Published:** 2023-07-26

**Authors:** Andrea Mihajlović, David Ivanov, Borislav Tapavički, Milica Marković, Dragana Vukas, Ana Miljković, Dejana Bajić, Isidora Semnic, Maja Bogdan, Dea Karaba Jakovljević, Stanislava Nikolić, Danijel Slavić, Dajana Lendak

**Affiliations:** 1Department of Physiology, Faculty of Medicine, University of Novi Sad, Hajduk Veljkova 3, 21137 Novi Sad, Serbia; 2Health Centre Novi Sad, Bulevar Cara Lazara 75, 21102 Novi Sad, Serbia; 3Department of General Medicine and Geriatrics, Faculty of Medicine Novi Sad, University of Novi Sad, Hajduk Veljkova 3, 21137 Novi Sad, Serbia; 4Department of Biochemistry, Faculty of Medicine Novi Sad, University of Novi Sad, Hajduk Veljkova 3, 21137 Novi Sad, Serbia; 5Faculty of Medicine, University of Novi Sad, Hajduk Veljkova 3, 21137 Novi Sad, Serbia; 6Clinic of Anesthesia and Intensive Care, University Clinical Center of Vojvodina, Hajduk Veljkova 1, 21137 Novi Sad, Serbia; 7Institute for Pulmonary Diseases of Vojvodina, Put Dr Goldmana Street 4, 21204 Sremska Kamenica, Serbia; 8Department of Pathophysiology and Laboratory Medicine, Faculty of Medicine, University of Novi Sad, Hajduk Veljkova 3, 21137 Novi Sad, Serbia; 9Center of Laboratory Medicine, Clinical Center of Vojvodina, Hajduk Veljkova 1, 21137 Novi Sad, Serbia; 10Department of Infectious Diseases, Faculty of Medicine, University of Novi Sad, Hajduk Veljkova 3, 21137 Novi Sad, Serbia; 11Clinic for Infectious Diseases, University Clinical Center of Vojvodina, Hajduk Veljkova 1, 21137 Novi Sad, Serbia

**Keywords:** COVID-19, SARS-CoV-2, biomarker, neutrophil–lymphocyte ratio, CRP, eosinophil count, prognosis

## Abstract

Various biomarkers like certain complete blood cell count parameters and the derived ratios including neutrophil–lymphocyte ratio are commonly used to evaluate disease severity. Our study aimed to establish if baseline levels of complete blood cell count-derived biomarkers and CRP, measured before any treatment which can interfere with their values, could serve as a predictor of development of pneumonia and the need for hospitalization requiring oxygen therapy. We retrospectively analyzed the laboratory data of 200 consecutive patients without comorbidities, who denied usage of medications prior to blood analysis and visited a COVID-19 ambulance between October and December 2021. Multivariate regression analysis extracted older age, elevated CRP and lower eosinophil count as significant independent predictors of pneumonia (*p* = 0.003, *p* = 0.000, *p* = 0.046, respectively). Independent predictors of hospitalization were higher CRP (*p* = 0.000) and lower platelet count (*p* = 0.005). There was no significant difference in the neutrophil–lymphocyte and platelet–lymphocyte ratios between examined groups. Individual biomarkers such as platelet and eosinophil count might be better in predicting the severity of COVID-19 than the neutrophil–lymphocyte and platelet–lymphocyte ratios.

## 1. Introduction

One of the main attributes of COVID-19 infections is their highly unpredictable clinical course. This is the primary reason why many researchers around the world have tried to find a reliable and easily obtainable biomarker that could provide an early warning sign against a potentially more severe case of COVID-19. Among the researched biomarkers, common candidates are the C-reactive protein (CRP) and many complete blood cell count (CBC) parameters, such as the neutrophil (NEU#), lymphocyte (LY#), eosinophil (EOS#), basophil (BAS#), monocyte (MONO#) and platelet (PLT) counts, but also certain indices derived from CBC parameters like the neutrophil/lymphocyte ratio (NLR), platelet/lymphocyte ratio (PLR) and lymphocyte/monocyte ratio (LMR). Interleukins, calprotectin and oxidative stress markers proved to have a good predictive value but these are seldom used prior to hospitalization as most primary care physicians do not have access to these tests [1,2,3]. As the CRP, CBC parameters and their derived ratios are the staples of a typical laboratory workup of patients suspected of having an infection, they could potentially be more useful in predicting the outcome of the disease. It has been shown that CRP, neutrophil count, aspartate aminotransferase, creatinine, LDH, D-dimer level and proinflammatory cytokines are lower in mild compared to severe cases [4]. Another study had shown that lymphopenia worsens with time and that the incidence of lymphopenia is higher in ICU patients compared to non-ICU patients [5]. A comprehensive meta-analysis had shown that the severe or fatal COVID-19 cases were characterized by elevated levels of CRP, ferritin and interleukins while the lymphocyte, eosinophil and platelet counts were decreased [6]. The same meta-analysis found that the elevated WBC happened due to neutrophilia while the lower WBC was usually due to a decrease in lymphocyte, monocyte and eosinophil counts. A study by Qin et al. pointed to lower lymphocyte count in severe cases, mainly due to the influence of coronavirus on T lymphocytes [7]. Also, an increase in eosinophil count could be an indicator of a more favorable outcome of COVID-19 [8]. In many diseases where systemic inflammation or stress is the underlying mechanism, it is not uncommon to find a combination of neutrophilia and lymphocytopenia [9]. This observation has led to the usage of the NLR as a biomarker of disease severity. Other biomarkers have also been used, such as the PLR and LMR [10,11], however, NLR remains the most frequently used prognostic biomarker in infective diseases [12,13]. The problem with interpreting the results of these studies is the values of these biomarkers are highly dependent on age, sex, comorbidities and certain drugs [10,14,15]. One study found that age, white blood cell (WBC) count and NLR are good predictors of mortality in COVID-19 patients but they did not consider other factors mentioned previously [16]. Another study pointed out the importance of NLR which was elevated in COVID-19 patients compared to healthy controls and even higher in ICU COVID-19 patients [17]. It is well known that corticosteroid therapy may lead to an increase in WBC count with a high NEU# and low LY#, EOS# and BAS# [18,19,20,21,22]. This raises the question of whether the COVID-19-positive patients were on corticosteroid therapy at the time many of these studies were conducted and, if they were, what kind of dosing regimen was used. Furthermore, macrolide antibiotics were also prescribed to COVID-19 patients and their anti-inflammatory and immunomodulatory effects are well documented and could interfere with the interpretation of the results [23,24]. A commentary by Bedel et al. also highlights the importance of the relative timing between blood sampling and disease onset, especially considering the potential impact that the various drugs taken during this time period can have on NLR, PLR and LMR values [25]. Regardless of its limitations, it was recommended that SARS-CoV-2-positive patients with NLR >6.11 should be started on corticosteroid therapy, which reflects the amount of importance given to this biomarker during the pandemic [26]. Along with NLR, PLR was also found to be an independent predictor of systemic inflammatory disease outcome, especially having in mind that both low LY# [27] and low PLT [28] on their own might indicate a worse outcome [29,30]. In COVID-19 patients suffering from chronic kidney disease (CKD), monocyte to lymphocyte ratio was a significant predictor of mortality among hospitalized patients and could potentially contribute to the early risk assessment in this patient population [31]. Also, both low EOS# [32,33] and low BAS# [34] were found to be potentially good predictors of the severe form of COVID-19.

The aim of this study was to:

(1) examine whether age, CRP, WBC, LY#, EOS#, BAS#, MONO#, PLT and their derived ratios (NLR, PLR and LMR) measured in the early stages of COVID-19 before any treatment were different between groups of: (a) patients without pneumonia vs. patients with chest X-ray-verified pneumonia and (b) non-hospitalized patients vs. hospitalized pneumonia patients requiring oxygen therapy; 

(2) determine which of these biomarkers have the biggest contribution in discriminating between the groups of: (a) patients without pneumonia vs. patients with chest X-ray-verified pneumonia and (b) non-hospitalized patients vs. hospitalized pneumonia patients requiring oxygen therapy.

## 2. Materials and Methods

This retrospective, single-center study included 200 consecutive patients, male and female, from the Health Centre Novi Sad (COVID-19 ambulance of primary healthcare level) who denied usage of any medications prior to blood laboratory analysis. Age of the participants varied from 20 to 81. Participants visited the COVID-19 ambulance due to experiencing symptoms of respiratory infection during the delta variant dominance, between October and December 2021. SARS-CoV-2 testing was performed upon their visit to the Health Centre Novi Sad using the nasopharyngeal swab. 

Participants were included in the study if they: (1) had a positive SARS-CoV-2 test (antigenic or PCR test—*Biomeireux SARS-CoV-2 R-GENE^®®^ Real Time Detection kit*) [35,36] and (2) visited the COVID-19 ambulance in the first 7 days from the onset of symptoms where they had their first laboratory tests prior to any treatment. Exclusion criteria were: (1) self-treatment prior to coming to the COVID-19 ambulance; (2) pregnancy and/or lactation, (3) presence of concomitant diseases, (4) patients on mechanical ventilation and (5) deceased. Participants underwent a physical exam, oximetry (SpO_2_) measurement, laboratory analyses (CBC and CRP were taken on the same day) and a chest X-ray based on which they were triaged for inpatient or outpatient care and followed by their initial clinic physician until the end of treatment. Patients with pneumonia had a chest X-ray to verify pneumonia. All hospitalized patients had SpO_2_ < 94% in room air and/or a respiratory rate of >30 per minute along with radiological evidence of pneumonia, therefore they belonged to the category of severe illness [37].

Hematological parameters (CBC, 5-part differential WBC and CRP) were measured from peripheral blood samples by the Mindray BC-5310 hematology analyzer (Shenzhen, China) before the start of treatment. All of the CBC parameters are expressed as number of cells per liter of blood. Several CBC parameters (for example, PLT count) were determined using the direct current (DC) impedance method. CRP values (in mg/L) were determined using a turbidimetric immunoassay. Differential WBC count was determined using flow cytometry.

This research was approved by the Ethics Committee of the Primary Health Centre in Novi Sad (decision number: 21/1).

### Statistical Analysis

Data were analyzed using SPSS v. 23.0 software. The normality of the data was assessed using the skewness, kurtosis and Kolmogorov–Smirnov tests. All parameters were represented as median ± interquartile range (IQR) as appropriate according to the distribution. Non-parametric approaches were used, since most continuous variables were skewed. Differences between groups were evaluated using the chi-square (χ2) test for categorical variables and the Mann–Whitney U test for continuous variables. A *p*-value < 0.05 was considered to be statistically significant. Binary logistic regression was performed to find independent predictors of pneumonia and hospitalization. In the regression model, we included all parameters that differed significantly between observed groups in univariate analysis. In the case of multicollinearity, for inclusion in multivariate analysis, we chose the factor with better results in univariate analysis. For some variables that were statistically significant, we constructed receiver operating characteristic (ROC) curves and measured areas under the curve (AUCs). 

## 3. Results 

Of the 200 participants in this study, 104 (52%) were male and 96 (48%) were female. The youngest participant was 20 and the eldest 81 years old. We divided our study sample into two groups based on the: (1) development of pneumonia and (2) need for hospital treatment requiring oxygen therapy. Of the 200 participants in the study sample, 124 (62%) had developed pneumonia while 76 (38%) had not developed pneumonia. On the other hand, 52 participants (26%) required hospitalization and oxygen therapy while 148 (74%) did not require them. 

### 3.1. Pneumonia

Age, CRP (*p* ≤ 0.001) and neutrophils (*p* = 0.031) were significantly higher in the group of patients with pneumonia compared to patients without pneumonia (Table 1, Figure 1a). Meanwhile, the eosinophil (*p* < 0.001) and platelet (*p* = 0.033) counts were significantly lower in group of patients with pneumonia (Table 1, Figure 1b,c). On the other hand, none of the derived ratios (NLR, PLR and LMR; *p* = 0.164, *p* = 0.134 and *p* = 0.102, respectively) was significantly different between the two groups (Table 1). A chi-square test revealed no statistically substantial connection between the development of pneumonia and the patients’ demographic data (sex) (*p* = 0.218).

In a multivariate logistic regression, the age (OR = 1.034, 95% CI: 1.012–1.057, *p* = 0.003), CRP (OR = 1.051, 95% CI: 1.025–1.077, *p* = 0.000) and eosinophil count (OR = 0.026, 95% CI: 0.001–0.931, *p* = 0.046) were significantly associated with the development of pneumonia (Table 2).

### 3.2. Hospitalization

When comparing hospitalized pneumonia patients with non-hospitalized patients, plasma CRP level (*p* < 0.001) and platelet (*p* = 0.006) and eosinophil (*p* = 0.032) counts showed statistically significant differences (Table 3, Figure 2a–c). We did not find significant differences in any other blood cell count parameters or calculated ratios. The chi-square test showed no significant connection between the hospitalization and the patients’ gender (*p* = 0.838).

**Multivariate logistic regression** analysis showed that higher CRP (OR = 1.021, 95% CI: 1.008–1.033, *p* = 0.000) and lower platelet count (OR = 0.989, 95% CI: 0.981–0.997, *p* = 0.005) were independent risk factors for the severe form of COVID-19, with a need for hospitalization and oxygen therapy later on (Table 4).

Although the platelet count of hospitalized patients was in the reference range, linear regression showed that this parameter is an independent predictor of hospitalization. We constructed a ROC curve in order to find a cut-off value of platelets that could indicate a greater risk of developing a more severe form of infection requiring oxygen therapy. ROC curve analysis (AUC ROC = 0.629, 95% CI 0.538–0.721, *p* = 0.006) shows that we can consider the platelet count of 180 × 10^9^/L as the cut-off value, with a sensitivity of 60% and a specificity of 58% (Figure 3).

## 4. Discussion

In our study, both the patients who developed pneumonia and those who required hospital treatment with oxygen therapy had significantly lower values of platelet and eosinophil counts and higher values of CRP, which is in line with previous studies by other authors [38,39]. None of the derived ratios (NLR, PLR and LMR) were significantly different, both when comparing patients with pneumonia to patients without pneumonia and when comparing hospitalized pneumonia patients to non-hospitalized patients.

An interesting finding in our study is that patients with pneumonia and those requiring oxygen therapy had significantly lower EOS#. Several previous studies [40,41,42] list eosinopenia as a characteristic finding in COVID-19-positive patients. Another study [43] found EOS# to be a significant element in the triage of COVID-19-positive patients, together with CRP. The importance of EOS# is also reflected in the discriminant analysis in our study (Table 4), as it was the third highest contributor in discriminating between hospitalized and non-hospitalized patients, second only to CRP and PLT. A study by Cortés-Vieyra and colleagues [33] found significant differences in EOS# and BAS# between deceased and recovered COVID-19 patients, with significantly lower values in the deceased group. Furthermore, in a study [44] including 85 cases of death from COVID-19, eosinopenia was a potential predictor of a negative outcome. Coming from a different angle, a couple of studies [45,46] have found eosinophilia to be a protective factor in COVID-19 infection. A study by Silva et al. [47] reported that EOS# was significantly lower in infected vs. non-infected patients and even found the cut-off value of 69 cells/μL to be the most appropriate predictor of infection. Furthermore, numerous studies pointed to the importance of eosinophils in respiratory infections caused by viruses such as RSV, influenza and parainfluenza [48]. Some of the mechanisms by which eosinophils are involved in the abovementioned infections are IFN-β production, stimulation of CD8+ cell response, nitric oxide (NO) production, accelerated virus clearance and limited viral replication [48]. Studies supposed that eosinopenia in infectious diseases develops due to several different mechanisms [49,50]. One of the previously considered potential mechanisms was the secretion of adrenal glucocorticoids and epinephrine as a part of the stress response during an acute infection [49]. Further research rejected the endocrine mechanism and proposed chemotactic substances such as the C5a of the complement system as the cause of transitory granulocytopenia followed by the eosinopenic–neutrophilic response [50]. Andreozzi et al. [51] pointed out that eosinopenia was commonly seen in both COVID-19 and influenza infections and that eosinopenia in these cases was usually accompanied by basopenia and monocytopenia.

Neutrophil count was significantly higher in the group of patients with pneumonia compared to the group of patients without pneumonia. On the other side, hospitalized patients had higher values of NEU# compared to non-hospitalized patients but this difference was not significant. The study by Masso-Silva et al. [52] pointed to an increase in neutrophil-associated cytokines and increased neutrophil count in the peripheral blood and lungs of COVID-19 patients. Another review found that neutrophilia and lymphocytopenia could be potential hematological markers present in COVID-19 infections [53]. On the other hand, the authors of a case report described severe neutropenia in COVID-19 and pointed out the importance of further investigation into this phenomenon and its cause [54]. Another study concluded that both high and low baseline neutrophil counts correlated with the fatal outcome, even though the mechanism by which COVID-19 causes neutropenia is unknown [55]. In the same study, the baseline neutrophil count was better at predicting mortality than the neutrophil count change rate, with the safe range of 1.64–4.0 × 10^9^/L [55]. Median values of patients with pneumonia and hospitalized patients in our study are inside this safe range, which fits the findings of the mentioned study as fatal cases of COVID-19 were not included in our study.

When comparing patients with and without pneumonia and hospitalized and non-hospitalized patients, the WBC and lymphocyte, monocyte and basophile counts did not significantly differ. In previous research, the lymphocyte count stood out as a significant predictor of severe COVID-19 [5,6,56]. Also, lymphopenia correlated well with the need for mechanical ventilation and fatal outcome [56]. However, our participants did not require mechanical ventilation and did not have a fatal outcome, so they all recovered. A meta-analysis also found that lymphopenia is associated with severe and critical (need for ICU treatment and acute respiratory distress syndrome) COVID-19 and increased mortality [57]. Another study [58] also found a good correlation between lymphopenia and mortality but did not find the lymphocyte count to be significantly different when comparing moderate and severe COVID-19 cases. Since our sample did not include critically ill COVID-19 patients, this could explain why we did not find significant differences in lymphocyte count.

We found that PLT was significantly lower in patients with pneumonia and there was a significant difference in PLT between non-hospitalized and hospitalized patients. Also, lower platelet count was an independent risk factor for hospitalization and oxygen therapy later on. A study by Boccatonda et al. [59] found that PLT was lower in COVID-19 patients compared to healthy controls but also made a broader conclusion that low PLT correlated with impaired blood gas analysis parameters and the outcome of the disease. The same authors stated that the PLT value obtained during the initial exam could be a good prognostic indicator of mortality, similar to our finding that initial PLT values could be used to discriminate between hospitalized and non-hospitalized patients. Another study [60] found that low PLT or a large drop from initial PLT values is a bad prognostic sign of ICU survival, while the study by Manne et al. [61] had shown that COVID-19 patients exhibited platelet hyperactivity with faster platelet aggregation. Significant alteration of platelet transcriptome and proteome in COVID-19 patients compared with healthy donors could be responsible for enhanced platelet aggregation, suggesting increased MAPK pathway activation as also partly responsible for pathological aggregation [61]. The role of platelets and the impact of thrombocytopenia in other viral infections are well known from previous studies [62,63] so these findings just underscore their importance in COVID-19.

The derived indices NLR, PLR and LMR were not significantly different in our study when comparing patients with pneumonia and patients without pneumonia but also when comparing hospitalized and non-hospitalized patients. A study by Mousavi-Nasab et al. [38] found higher CRP, WBC and NEU# in patients with a more severe form of COVID-19 just like in our study, but they also found the NLR to be significantly higher in this group of patients, which is not the case in our study. This can be partially explained by the fact that the NLR is influenced by sex, age and ethnicity [64]. For example, in the USA, the average NLR in a healthy population was 2.24 among Caucasians and 1.76 among African Americans [65] while the average in central China was 1.72 [10]. A study by Haghjooy Javanmard et al. [66] controlled for age and sex and found that an NLR > 6.5 increased the chance of disease severity by 4 times. In our study, the NLR values were higher than normal values [14] in both patients with (3.01 ± 2.13) and without pneumonia (2.88 ± 2.68) and even higher in hospitalized pneumonia patients (3.27 ± 2.27). A study by Toori et al. [67] found that the NLR was significantly higher in severe COVID-19 cases. Another study [68] also reported higher NLR values in severe forms of COVID-19 and characterized it as a useful prognostic factor in detecting severe cases. On the other hand, a separate study [69] found that an NLR higher than 3.13, a value lower than the one in our study, was the predictor of a bad outcome but only in patients who are over 50 years old. This could indicate the need to take more different parameters into consideration when interpreting the NLR in order to predict the outcome of the disease.

The other two ratios used in our study, PLR and LMR, were not significantly different either between patients with and without pneumonia or between hospitalized and non-hospitalized patients. In fact, the PLR in our study was higher in patients who did not develop pneumonia and in non-hospitalized patients which is contrary to previous studies [70]. Namely, the PLR is usually elevated and is thought to be indicative of increased platelet activity seen in inflammatory conditions, such as viral infections and cardiovascular disease [71]. It is considered that an increase in PLR during COVID-19 is due to a more pronounced drop in LY# relative to PLT, while in other conditions it might be due to an increase in PLT [72]. In our study, patients without pneumonia had the highest PLR (144 (98-208)) while the hospitalized pneumonia patients had a lower PLR (110.7 (86.8–173.0)). We looked at the parameters used to calculate this ratio—PLT and LY#—and noticed that the hospitalized pneumonia patients had a relatively larger decrease in PLT compared to the decrease in LY# which could explain the lower PLR values. A recent study [73] looking at COVID-19 patients that visited the emergency room reported they were not able to discriminate between the moderate and severe forms of the disease using the PLR. These results could be explained by the variable dynamics of CBC parameters, especially PLT and LY#, during the course of a COVID-19 infection. We would like to emphasize once again that the data in our study were obtained in the first 7 days of the infection before any treatment had begun and that we did not include deceased patients’ data but only the data from COVID-19 patients who recovered from the disease. Lastly, as the triaging of patients requires quick decision making from the physician, using a single parameter might prove to be more useful. On the other hand, developing a formula to calculate the risk of development of severe COVID-19 could be useful but would have to take into account different parameters such as patient age, comorbidities, sex and ethnicity.

## 5. Conclusions

Biomarkers such as CRP and individual CBC parameters (EOS# and PLT), measured before any treatment that could interfere with laboratory results, may be predictive of the severe form of COVID-19, with the need for hospitalization and oxygen therapy later on. A cut-off value of platelets measured in the early stages of COVID-19 that could indicate a greater risk of developing a more severe form of infection requiring oxygen therapy could be 180 × 10^9^/L. Individual CBC parameters may be better as predictors of the development of pneumonia and also severe COVID-19 compared to CBC-derived ratios such as the NLR, PLR and LMR if other variables such as sex, age and ethnicity are not taken into account, which is not uncommon.

## Figures and Tables

**Figure 1 healthcare-11-02137-f001:**
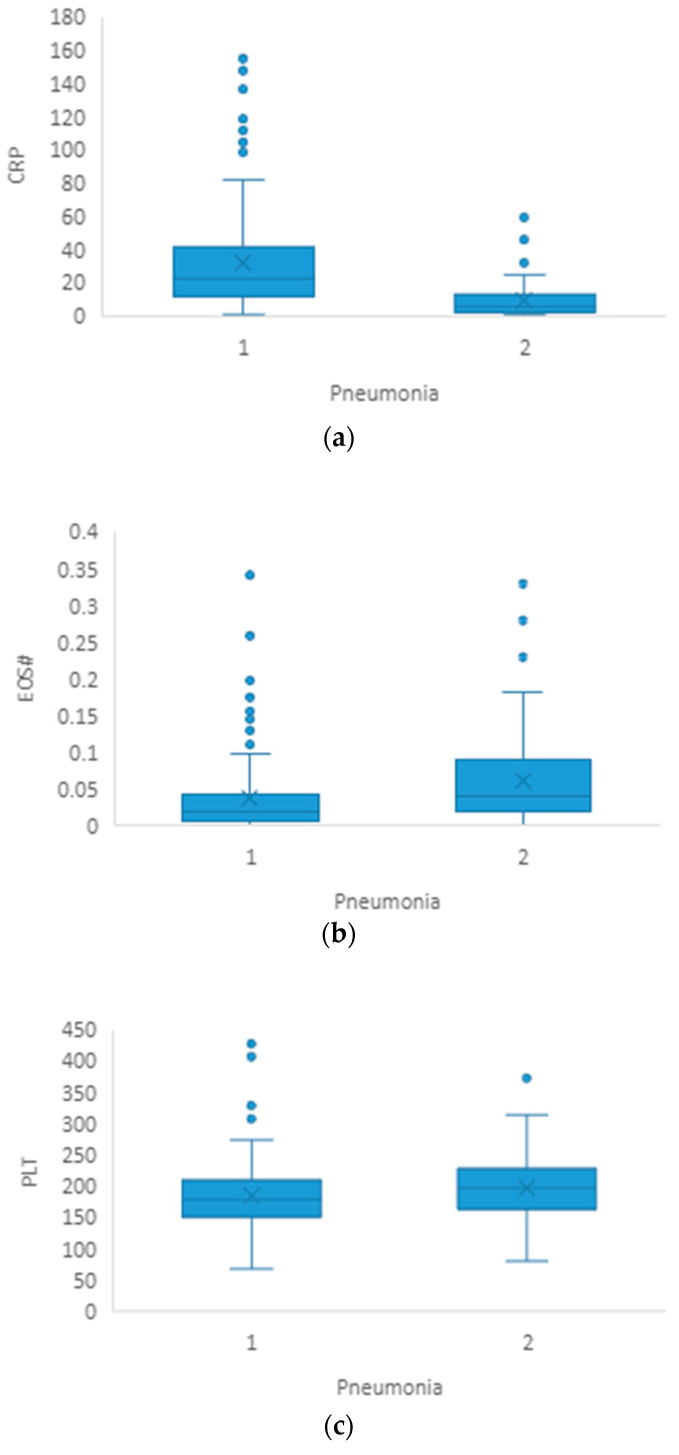
(**a**–**c**) Graphical representation of median and IQR for CRP, EOS# and PLT between groups of patients with pneumonia (1) and without pneumonia (2).

**Figure 2 healthcare-11-02137-f002:**
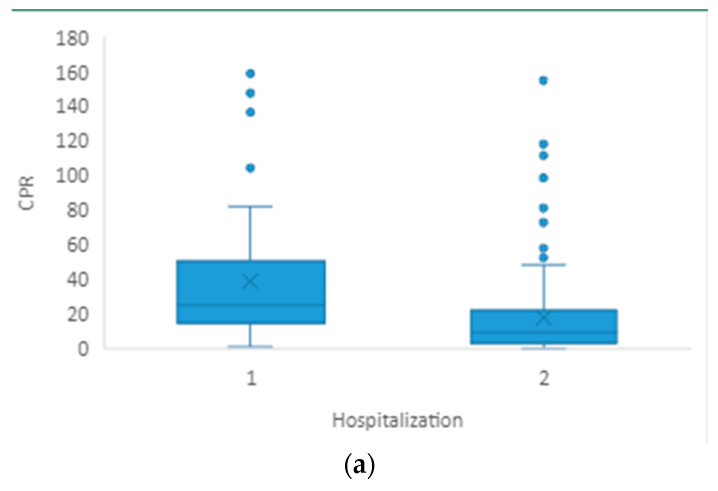
(**a**–**c**) Graphical representation of median and IQR for CRP, EOS# and PLT between groups of patients requiring hospitalization (1) and patients not requiring hospitalization (2).

**Figure 3 healthcare-11-02137-f003:**
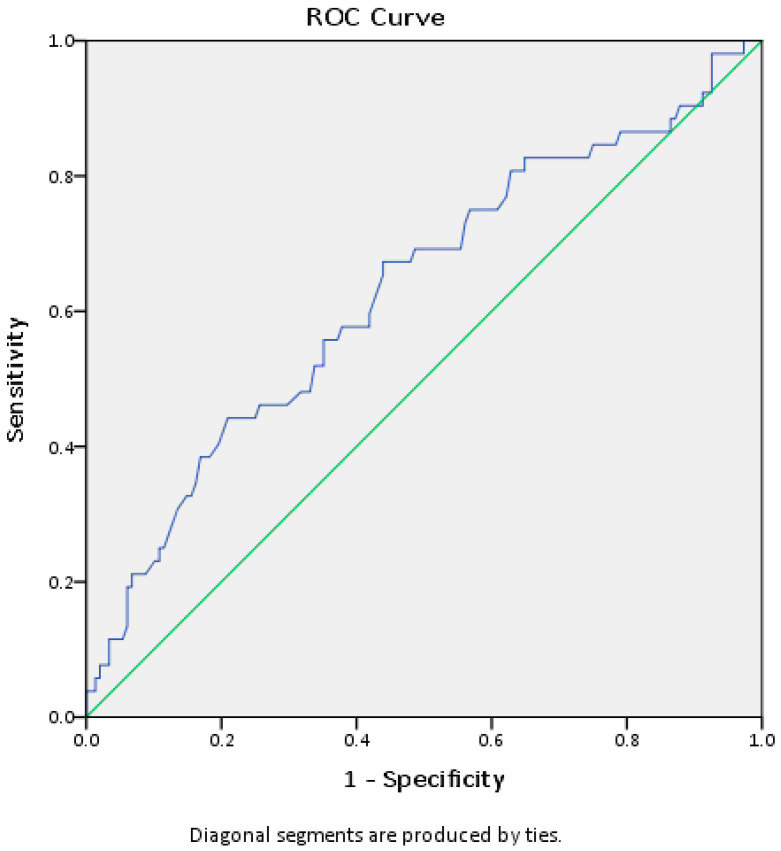
ROC curve of platelet count for COVID-19 patient outcome.

**Table 1 healthcare-11-02137-t001:** Association of biomarkers’ values and the development of pneumonia (Mann–Whitney U test).

	Without Pneumonia (n = 76) Median (IQR)	Pneumonia (n = 124) Median (IQR)	*p*-Value
**Age**	48.5 (35.5–64.0)	58.0 (45.0–68.0)	0.001
**CRP** **(mg/L)**	6.3 (2.2–13.0)	22.5 (11.6–43.7)	<0.001
**WBC** **(×10^9^/L)**	4.70 (4.05–5.80)	5.15 (3.92–6.70)	0.082
**NEU#**	2.87 (2.17–4.01)	3.50 (2.34–4.80)	0.031
**LY#**	1.38 (0.94–1.90)	1.36 (1.08–1.81)	0.691
**MONO#**	0.40 (0.31–0.60)	0.38 (0.27–0.51)	0.065
**EOS#**	0.04 (0.02–0.10)	0.02 (0.007–0.04)	<0.001
**BASO#**	0.012 (0.006–0.020)	0.010 (0.005–0.015)	0.087
**PLT** **(×10^9^/L)**	199 (164–227)	179 (152–211)	0.033
**NLR**	2.22 (1.22–3.48)	2.42 (1.60–3.68)	0.164
**PLR**	144 (98–208)	129 (92–172)	0.143
**LMR**	3.27 (2.10–5.12)	3.80 (2.51–5.44)	0.102

Values are expressed as median ± IQR. CRP: C-reactive protein; WBC: white blood cell count; NEU#: neutrophil count; LY#: lymphocyte count; MONO#: monocyte count; EOS#: eosinophil count; BASO#: basophil count; PLT: platelet count; NLR: neutrophil–lymphocyte ratio; PLR: platelet–lymphocyte ratio; LMR: lymphocyte–monocyte ratio.

**Table 2 healthcare-11-02137-t002:** Multivariate logistic regression for the prediction of pneumonia development.

	*p*	Odds Ratio	95% Confidence Interval for Odds Ratio	
			Lower	Upper
**Age**	0.003	1.034	1.012	1.057
**PLT**	0.206	0.996	0.991	1.002
**NEU#**	0.413	1.105	0.871	1.402
**EOS#**	0.046	0.026	0.001	0.931
**CRP**	0.000	1.051	1.025	1.077
**Constant**	0.052	0.196		

CRP: C-reactive protein; NEU#: neutrophil count; EOS#: eosinophil count; PLT: platelet count.

**Table 3 healthcare-11-02137-t003:** Association of biomarkers’ values and the need for hospitalization (Mann–Whitney U test).

	Hosp (n = 52) Median (IQR)	NoHosp (n = 148) Median (IQR)	*p*-Value
**Age**	60 (45–69)	52 (41–66)	0.084
**CRP** **(mg/L)**	28.6 (15.3–57.7)	10.4 (3.0–22.8)	<0.001
**WBC** **(×10^9^/L)**	5.50 (4.25–7.05)	4.90 (3.90–5.95)	0.054
**NEU#**	3.62 (2.65–5.12)	3.03 (2.25–4.26)	0.051
**LY#**	1.33 (1.01–1.81)	1.38 (1.08–1.83)	0.905
**MONO#**	0.36 (0.29–0.53)	0.39 (0.29–0.55)	0.526
**EOS#**	0.018 (0.006–0.047)	0.030 (0.011–0.067)	0.032
**BASO#**	0.010 (0.005–0.179)	0.011 (0.006–0.020)	0.251
**PLT** **(×10^9^/L)**	167 (141–204)	195 (160–223)	0.006
**NLR**	2.76 (1.61–3.73)	2.21 (1.35–3.37)	0.081
**PLR**	110.7 (86.8–173.0)	139.5 (99.3–194.6)	0.061
**LMR**	3.81 (2.43–5.28)	3.48 (2.38–5.30)	0.624

Values are expressed as median ± IQR. Hosp: patients requiring hospitalization; NoHosp: patients not requiring hospitalization; CRP: C-reactive protein; WBC: white blood cell count; NEU#: neutrophil count; LY#: lymphocyte count; MONO#: monocyte count; EOS#: eosinophil count; BASO#: basophil count; PLT: platelet count; NLR: neutrophil–lymphocyte ratio; PLR: platelet–lymphocyte ratio; LMR: lymphocyte–monocyte ratio.

**Table 4 healthcare-11-02137-t004:** Multivariate logistic regression for the prediction of hospitalization requiring oxygen therapy.

	*p*	Odds Ratio	95% Confidence Interval for Odds Ratio	
			Lower	Upper
**PLT**	0.005	0.989	0.981	0.997
**NEU#**	0.356	1.120	0.880	1.426
**EOS#**	0.162	0.003	0.000	10.771
**CRP**	0.001	1.021	1.008	1.033
**Constant**	0.736	1.265		

CRP: C-reactive protein; WBC: white blood cell count; NEU#: neutrophil count; EOS#: eosinophil count.

## Data Availability

Data are unavailable due to privacy and ethical restrictions.

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
