# Peer review of "Prognostic Value of Routine Biomarkers in the Early Stage of COVID-19"

_healthcare, 2023, doi:10.3390/healthcare11152137_

Round 1

Reviewer 1 Report

Mihajlovic et al presented a study of routine biomarkers in the early stages of COVID-19.

Even though the study is based on statistics from 200 COVID-19 patients, it gives an idea of the biomarkers to be measured. 

The authors show that individual CBC parameters may be better compared to the CBC-derived ratios such as NLR, PLR, and NMR. This clinical data builds a perspective to diagnose patients at the early stages of COVID-19.

My only suggestion would be to try and demonstrate the numbers in graphs, if possible, along with the table.

Thank you.

The data presented in the manuscript is worth publishing as it gives a new  perspective for the diagnosis of COVID-19.

Reviewer 2 Report

The authors had done a study with 200 COVID-19 patients with the CBC. This will give a significant impact on clinical diagnosis. 

However, in title 
"Prognostic value ... early stage of COVID-19" 
Based on CDC release in March 2023: The mean incubation period was 2.1 to 3 days only for those with omicron. In which, 7 days of patients used in this study was no longer defined as early

In introduction, 
-detail explanation was given about the CBC to inflammation, however, it should be more specifically to SARS-COV-2. 
-the introduction  should highlight on prognostic of using CBC in COVID-19 but not "COVID-19 on clinical therapy, e.g. cortisoid therapy "

Materials & Methods, 
study showed using the standard gold methods for COVID-19 detection using either Ag or PCR test. However, it is unclear the threshold cycle of the PCR test. Besides, it is unclear how the study diagnosed patients as pneumonia (is that diagnosed through an x-ray by the clinical physician or any complementary test)
Line 113 - give DC in full 
Line 113 - CRP was done (however this is not clear whether CRP was done as on the same day of Full blood count or a different day)  

3.Results 
The result relates the CBC with pneumonia and hospitalization status only. There is no analysis on CBC with the signs and symptoms that able to classify the COVID-19 into mild , moderate or severe. 

4. Discussion 
The study discusses the findings with low platelets, eosinophils, and CRP.

However,

in line 212-214, it will be great to further explain the mechanism of why eosinophils are always being migrated but not neutrophils. 

Line 226-228, why the involvement of platelet in aggregation n COVID-19 

Discussion on why lymphocytes and other cells count was not affect seriously 

Reviewer 3 Report

There needs to be more study to assist the investigation conducted in this study. 

-- The laboratory experiments done are not enough to make the conclusion that was made.

-- The discussion part needs more elaboration in light of the current literature.

-- It would be ideal to show additional parameters and their relevance to the study conducted.

-- It would be better to express the data to show the relationship between the CBC parameters and their relevance to different diseases.

Reviewer 4 Report

This paper “Prognostic Value of Routine Biomarkers in the Early Stage of  COVID-19” by Andrea Mihajlović et al has reported an interesting finding  that patients with pneumonia and those requiring oxygen therapy had significantly lower EOS# and the analysis also showed that it’s a great factor deciding hospitalization.

Suggestion:

The authors can improve study by adding a table showing oxygen saturation (SpO2) level of both hospitalized and non-hospitalized patients.

Round 2

Reviewer 2 Report

The author made the appropriate revision based on the suggestion. 

Reviewer 3 Report

Commments were addressed properly